# Comparison of two one-piece acrylic foldable intraocular lenses: Short-term change in axial movement after cataract surgery and its effect on refraction

**So Goto**[1,2,3], **Naoyuki Maeda**[1]*, **Kazuhiko Ohnuma**[4], **Toru Noda**[2]

**1** Department of Ophthalmology, Osaka University Graduate School of Medicine, Osaka, Japan, **2** National Hospital Organization, Tokyo Medical Center, Tokyo, Japan, **3** Herbert Wertheim School of Optometry and Vision Science, University of California, Berkeley, Berkeley, California, United States of America, **4** Laboratorio de Lente Verde, Sodegaura, Chiba, Japan

* nmaeda@ophthal.med.osaka-u.ac.jp

**Data Availability Statement:** All relevant data are within the paper and its Supporting information files.

## Abstract

### Purpose

To compare the change in intraocular lens (IOL) axial movement, corneal power, and post-operative refraction of eyes implanted with two different single-piece, open loop, acrylic foldable IOLs with planar-haptic design: one IOL with hinges vs. one IOL without hinges. The role of IOL axial movement on short-term refractive shift after cataract surgery was also evaluated.

### Methods

This retrospective comparative study enrolled consecutive patients who had phacoemulsification with aspheric IOL implantation. The IOL depth (the distance from corneal endothelium to IOL surface) and corneal power were measured via anterior-segment optical coherence tomography at 4 days and 1 month postoperatively. The changes in axial movement of the IOL, corneal power, and manifest refractive spherical equivalent (MRSE) were compared among groups, and the correlations between each lens were evaluated.

### Results

IOL with hinges was implanted in 42 eyes of 42 patients and IOL without hinges was implanted in 42 eyes of 42 patients. The change in axial movement between 4 days and 1 month was significantly smaller in the IOL with hinges group than in the IOL without hinges group ($p < 0.001$). The axial movement of IOL with hinges did not correlate with the MRSE change; however, the forward shift of IOL without hinges correlated with the myopic refractive change (Pearson $r = 0.62$, $p < 0.001$).

### Conclusion

The postoperative axial movement of IOL was more stable in the IOL with hinges group than the IOL without hinges group between 4 days and 1 month after cataract surgery. Even

**Funding:** The editorial assistance in the preparation of the manuscript was performed with funding from Alcon Japan Ltd (IIT # 54635107). The funders had no role in study design, data collection and analysis, decision to publish, or preparation of the manuscript.

**Competing interests:** SG is supported by Grants Manpei Suzuki Diabetic Foundation and Japan Society for the Promotion of Science Overseas Research Fellowships that are not related to the submitted work. SG, NM, and TN received honoraria for speaking outside the submitted work from Alcon Japan Ltd. These do not alter our adherence to PLOS ONE policies on sharing data and materials. There are no patents, products in development or marketed products to declare.

though the two study IOLs with planar-haptic design are made of similar acrylic materials, other characteristics such as hinge structure may affect IOL stability in the bag.

## Introduction

Postoperative intraocular lens (IOL) position can affect postoperative refraction, and the gap between actual and target refraction remains a major concern in cataract surgery [1, 2]. IOL characteristics play an important role in determining postoperative lens position. Although previous studies have compared postoperative axial movement of single- and three-piece IOLs, with the single-piece IOLs having less axial movement and resulting in more stable refraction [3–8], short-term myopic changes in refraction from one day to one month after cataract surgery with implantation of single-piece acrylic IOLs were recently reported [9, 10]. Despite the substantial number of IOL models on the market and the popularity of single-piece acrylic IOLs, little is known about differences in postoperative axial movement between one-piece IOLs with planar-haptics from different manufacturers and its effect on the postoperative refraction.

In vitro compression assessments have demonstrated that there is a difference in the axial movement of the IOL optics when one-piece IOLs are designed with planar haptics [11–14]. Hence, we hypothesized that the hinge design, which may affect the axial movement of the IOL optics, is partially responsible for stability of the IOL position.

The purpose of this study was to compare the short-term axial movements of two types of IOLs using anterior-segment optical coherence tomography (AS-OCT). Both IOLs have a similar material (hydrophobic acrylic), 1-piece open loop, and planar haptics; however, both lenses have different haptics design. In addition, we evaluated the role of IOL position shift on refractive change after cataract surgery.

## Methods

This was a retrospective, observational, comparative, single-center study of all patients who had undergone uncomplicated cataract surgery at the National Hospital Organization, Tokyo Medical Center, Tokyo, Japan, between December 2015 and February 2017. The study was approved by the institutional review board of the National Hospital Organization, Tokyo Medical Center; Tokyo, Japan, and was conducted in accordance with the tenets of the Declaration of Helsinki. Each patient provided written consent for their medical records to be used in this study.

Only eyes undergoing their first cataract surgery were included. Patients were excluded if they had previous ocular surgery, history of ocular trauma, presence of significant ocular comorbidities, unreliable preoperative biometric measurements, IOL implantation outside the capsular bag, dislocated IOL, intraoperative or postoperative complications, or corrected distance visual acuity (CDVA) after cataract surgery less than 20/30.

All surgical procedures were performed under topical anesthesia by the same experienced surgeon (TN). First, a continuous curvilinear capsulorrhexis measuring approximately 5.0 mm in diameter was accomplished using a bent needle. Subsequently, a clear cornea temporal self-sealing 2.2-mm incision was made, followed by phacoemulsification and in-the-bag unilateral or bilateral IOL implantation of either an IOL with hinges (AcrySof® IQ Toric IOL [Alcon Vision, LLC; Fort Worth, Texas]) or an IOL without hinges (Vivinex™ iSert® XY-1 IOL [Hoya Corporation; Tokyo, Japan]). Both of the IOLs have 13.0 mm overall diameter, 6.0 mm optic diameter, and planar haptics designs that have a 0-degree angle, whereas IQ Toric IOL has a

specific flexible hinge design (Fig 1). The same ophthalmic viscoelastic device (Opegan Hi, Santen Pharmaceutical, Osaka, Japan) was used for all surgeries. The I/A tip was inserted behind the IOL optic and the posterior chamber was directly irrigated and cleaned.

Preoperative axial length (AL), central corneal thickness (CCT), anterior chamber depth (ACD), lens thickness (LT), and keratometry (K) were measured using the swept-source OCT-based biometer OA-2000 (Tomey Corporation, Nagoya, Japan). The corneal real power including anterior and posterior corneal refractive powers, corneal thickness and IOL depth,

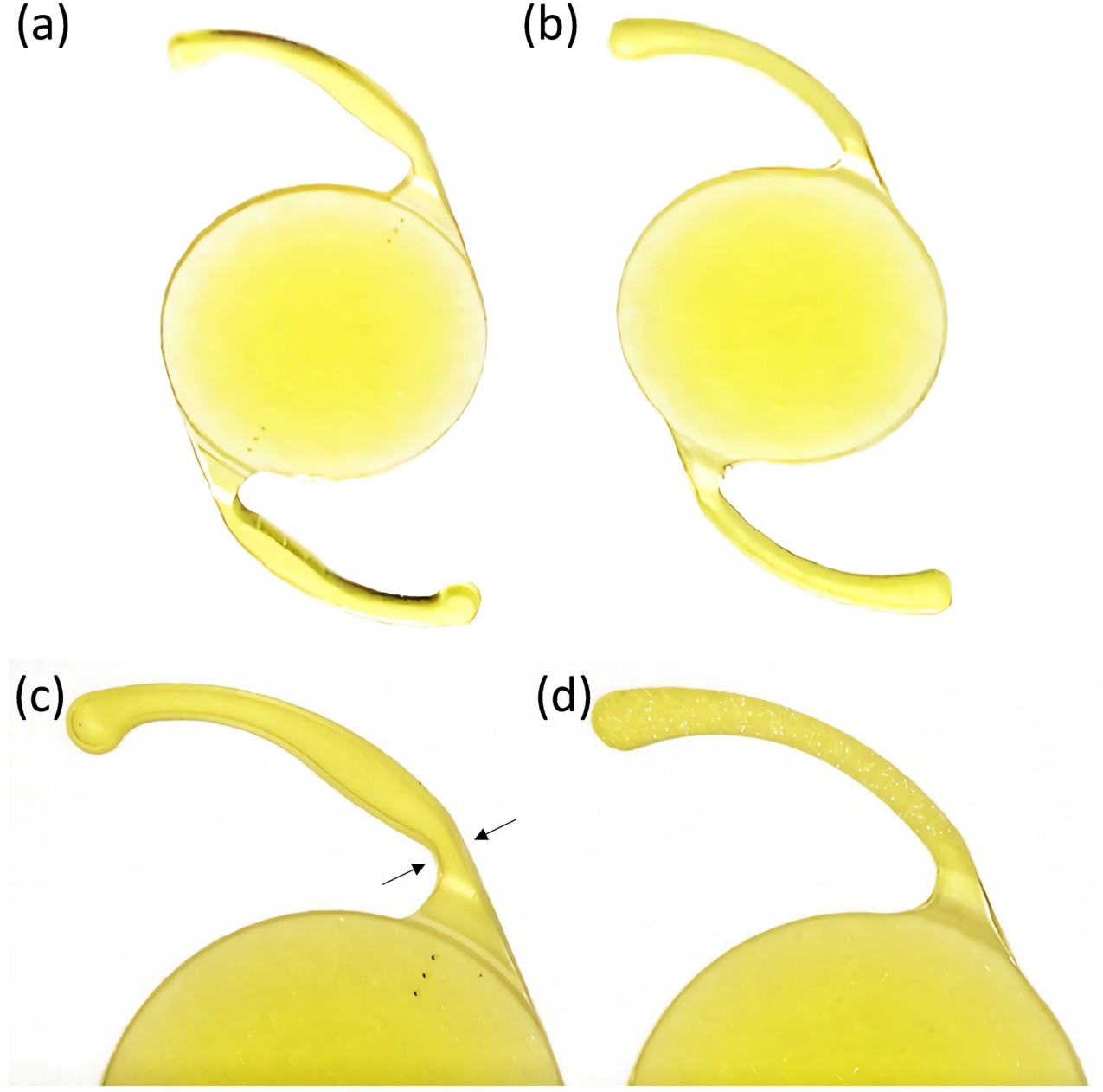

**Fig 1.** Optic and haptic configurations of (a) IOL with hinges and (b) IOL without hinges. While both of IOLs have planar haptics designs with a 0-degree angle, the haptics of the IOL with hinges (c) are constricted (i.e. flexible hinge design, black arrows) and the haptics of the IOL without hinges (d) are straight.

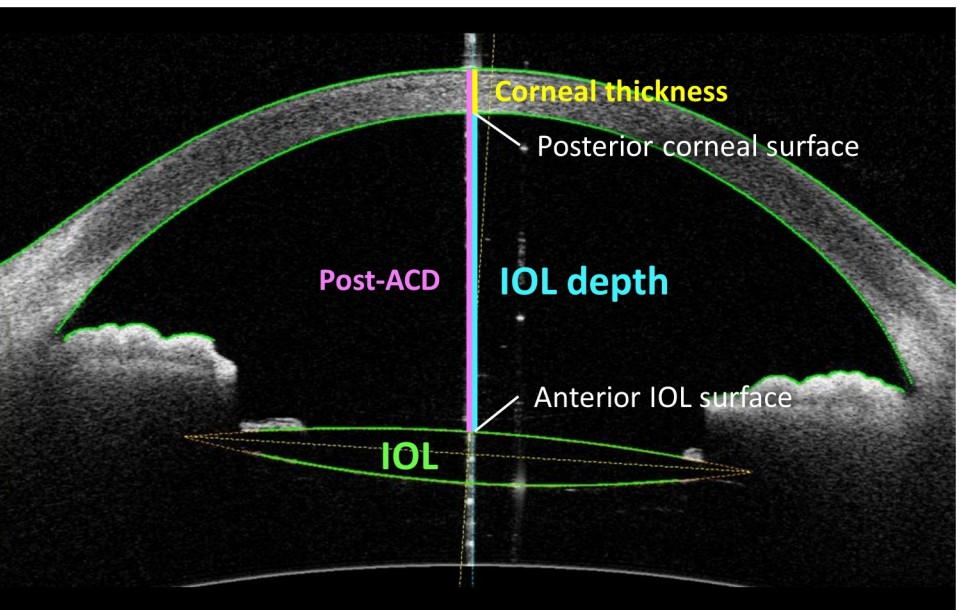

**Fig 2. IOL depth measured using anterior-segment optical coherence tomography.** IOL, intraocular lens, Post-ACD; postoperative anterior chamber depth.

which is the distance from corneal endothelium to IOL surface along the vertex normal (Fig 2), were automatically measured using a swept-source AS-OCT (CASIA2, SS-2000; Tomey Corp; Aichi, Japan) with a super-luminescent diode light source (1310 nm wavelength) and a scan speed of 50,000 A-scans/second. To precisely measure a true change in IOL position or in corneal curvature, three repeated measurements were obtained during a single visit by one technician. The exported data is the average value of three repeated measurements. The OCT images were obtained with dilated pupils. The estimated refractive error associated with the change in IOL depth was calculated using OpticStudio 16.5 Sp5 (Zemax, LLC.) [15, 16]. The calculations with OpticStudio were performed in the paraxial form. Therefore, the number of rays and aperture size did not contribute to the calculated results. Although asphericity affects the spherical results when comparing 4 days and 1 month spherical differences, to simplify the analysis, asphericity can be removed from the calculation of spherical differences. The eye model was built from the patients' biometry data including the cornea power, aqueous depth, and AL (refer to the S1 File).

Manifest refractive spherical equivalent value (MRSE) determined as the spherical power plus half the cylindrical power and AS-OCT were measured postoperatively at 4 days and 1 month. All examinations, including a CDVA measurement using a Landolt C chart at 5 meters, were performed by experienced ophthalmic technicians unaware of the purpose of the study.

Statistical analyses were performed with JMP Pro version 14.3.0 (SAS Institute Inc.). Normality of data distribution was assessed using the Shapiro-Wilk normality test. Differences between IOL groups in IOL depth and other continuous variables with normal distribution were compared with an unpaired t-test. Continuous variables without normal distribution were compared using the Mann-Whitney U test. The data between each time-interval pair were compared using the paired t-test or Wilcoxon signed-rank test. The Pearson correlation coefficient was used to determine the strength of the linear association between the changes of

MRSE and the changes in IOL depth and corneal power. The sample size was calculated to detect a difference in error of 0.1 mm between 2 groups; with a significance level of 5%, a statistical power of 80%, and assuming standard deviation (SD) to 0.11 mm, 41 eyes were required. Differences with a p-value less than 0.05 were considered statistically significant.

## Results

The IOL with hinges was implanted in 42 eyes of 42 patients, and IOL without hinges was implanted in 42 eyes of 42 patients. Table 1 summarizes the demographic data and ocular dimensions of the two IOL groups. The mean age, ratio of men to women, preoperative AL, CCT, ACD, LT, K, and implanted IOL power did not significantly differ between the groups.

Mean postoperative IOL depth and ACD over time is shown in Table 2. Compared with the IOL without hinges group, the IOL with hinges group had significantly less change in IOL depth and ACD from 4 days to 1 month ($p = 0.0002$ and $p = 0.0004$, respectively; Fig 3). Accordingly, the IOL with hinges group had significantly less absolute axial movement from 4 days to 1 month compared with the IOL without hinges group ($p = 0.0004$). While corneal thickness and postoperative ACD obtained by swept-source AS-OCT were significantly reduced from 4 days to 1 month in both group, there was no significant difference in the change in corneal thickness between the two groups. Although posterior corneal power was statistically significant different from 4 days to 1 month after surgery in both groups, postoperative total and anterior corneal powers of each group did not show significant changes (Table 3). The IOL without hinges group had a significant myopic shift in MRSE compared with the IOL with hinges group ($p = 0.03$, Table 4).

To investigate the effect of the change in IOL depth on postoperative refraction, a general linear model analysis was performed for each group (Fig 4). The refractive change related to IOL shift from 4 days to 1 month in the IOL without hinges group was significantly correlated with the change in MRSE from 4 days to 1 month ($r = 0.62$, 95% confidence interval [CI] of 0.39 to 0.78; $p < 0.0001$); whereas there was no significant correlation in the IOL with hinges group ($r = 0.22$, 95% CI of -0.09 to 0.49; $p = 0.15$).

The change in corneal power was not significantly correlated with the change in MRSE in both the IOL with hinges ($r = 0.24$, 95% CI of -0.07 to 0.50; $p = 0.13$) and IOL without hinges groups ($r = 0.05$, 95% CI of -0.26 to 0.35; $p = 0.77$).

**Table 1. Demographic data and preoperative optical properties of the eyes.**

| Characteristic | IOL with hinges group n = 42 | | IOL without hinges group n = 42 | | P Value |
|---|---|---|---|---|---|
| | Mean + SD | Range | Mean + SD | Range | |
| Age (years) | 73.8 ± 8.1 | 56 to 85 | 72.9 ± 7.8 | 56 to 87 | 0.57 |
| Male/Female | 14/28 | - - - | 16/26 | - - - | 0.65 |
| Axial length (mm) | 24.24 ± 1.64 | 21.68 to 28.58 | 23.90 ± 1.27 | 22.22 to 28.19 | 0.26 |
| Ksteep (D) | 44.92 ± 1.66 | 41.21 to 49.71 | 44.29 ± 1.31 | 40.61 to 47.14 | 0.06 |
| Kflat (D) | 43.69 ± 1.61 | 40.37 to 48.01 | 43.64 ± 1.30 | 40.27 to 46.68 | 0.89 |
| Corneal thickness (mm) | 525 ± 29 | 454 to 583 | 528 ± 29 | 467 to 594 | 0.67 |
| ACD (mm) | 3.17 ± 0.35 | 2.25 to 3.75 | 3.16 ± 0.32 | 2.46 to 4.01 | 0.60 |
| Lens thickness (mm) | 4.55 ± 0.41 | 3.60 to 5.26 | 4.54 ± 0.33 | 3.96 to 5.41 | 0.91 |
| IOL (D) | 20.1 ± 4.2 | 10 to 26 | 21.4 ± 3.0 | 13 to 26.5 | 0.12 |

ACD; anterior chamber depth, D; diopter, IOL; intraocular lens, K; corneal power, SD; standard deviation.

**Table 2. Comparison of mean (± SD) corneal thickness, postoperative anterior chamber depth, and intraocular lens depth, change in each parameter and absolute axial movement.**

| Parameters | IOL with hinges group n = 42 | | IOL without hinges group n = 42 | | P Value[a] |
|---|---|---|---|---|---|
| Corneal thickness (um) | Mean ± SD | Range | Mean ± SD | Range | |
| 4 days postop. | 562 ± 38 | 482 to 644 | 566 ± 30 | 510 to 647 | 0.59 |
| 1 month postop. | 544 ± 32 | 471 to 615 | 545 ± 31 | 486 to 636 | 0.91 |
| P Value[b] | < .0001*** | | < .0001*** | | |
| Postoperative ACD (mm) | | | | | |
| 4 days postop. | 4.74 ± 0.27 | 4.30 to 5.29 | 4.75 ± 0.27 | 4.32 to 5.75 | 0.89 |
| 1 month postop. | 4.70 ± 0.25 | 4.10 to 5.24 | 4.62 ± 0.25 | 4.21 to 5.53 | 0.15 |
| P Value[b] | 0.002** | | < .0001*** | | |
| IOL depth (mm) | | | | | |
| 4 days postop. | 4.19 ± 0.27 | 3.72 to 4.76 | 4.19 ± 0.28 | 3.73 to 5.24 | 0.95 |
| 1 month postop. | 4.17 ± 0.25 | 3.56 to 4.77 | 4.08 ± 0.26 | 3.63 to 5.04 | 0.1 |
| P Value[b] | 0.13 | | < .0001*** | | |
| Change in corneal thickness (um) | 17.8 ± 17.2 | -11 to 75 | 21.1 ± 12.2 | -2 to 51 | 0.31 |
| Change in postoperative ACD (mm) | 0.05 ± 0.09 | -0.11 to 0.24 | 0.13 ± 0.12 | -0.07 to 0.42 | 0.0004*** |
| Change in IOL depth (mm) | 0.02 ± 0.09 | -0.13 to 0.23 | 0.11 ± 0.12 | -0.10 to 0.40 | 0.0002*** |
| Refractive change associated with change in IOL depth (D)[c] | 0.02 ± 0.11 | -0.19 to 0.39 | 0.15 ± 0.16 | -0.15 to 0.55 | 0.0001*** |
| Absolute axial movement (mm) | 0.07 ± 0.06 | 0 to 0.23 | 0.13 ± 0.09 | 0.02 to 0.40 | 0.0004*** |

ACD = anterior chamber depth; IOL = intraocular lens; *Statistically significant difference, P < 0.05; P Value[a] between the 2 IOL groups; P Value[b] between the intervals; [c] calculated by the OpticStudio software.

## Discussion

The present study demonstrates that the mean postoperative IOL depth of the IOL with hinges group was stable during the first month from postoperative day 4, while the mean postoperative IOL depth of the IOL without hinges group significantly decreased during the same period. The change in postoperative IOL depth between day 4 and 1 month was significantly correlated with the change in MRSE in the IOL without hinges group.

According to a previous report, the postoperative IOL depth significantly decreased between day 1 and 1 month in eyes with an AcrySof SN60WF IOL, which has the same haptic design as Acrysof IQ Toric IOLs, AMO ZCB00V IOLs, and Hoya XY-1 IOLs [9]. Recently, Clareon CNA0T0 IOL(Alcon), which also has the same haptics design as the AcrySof IQ Toric IOL, was reported to have a significant anterior shift of postoperative IOL depth between day 1 and 1 month [10]. In the present study, we set the IOL depth at postoperative day 4 as the reference and compared that value with the IOL depth at 1 month postoperative. Additionally, AL was positively correlated with the change in IOL depth from 4 days to 1 month in only the IOL without hinges group (S1 Fig). As a consequence, the IOL depth of IOL with hinges was stable from postoperative day 4 to postoperative 1 month. Considering that the haptics of the AcrySof IQ Toric IOL have a specific flexible hinge design in which the axial stiffness is greater than the lateral stiffness [13] and the haptics of the Vivinex iSert XY-1 IOL are straight (Fig 1), the presence or absence of the hinge may cause differences in postoperative anterior shift of the IOL even if IOLs are designed with planar haptics.

Based on OpticStudio software, a 20 D IOL with a 24 mm AL, 7.7 mm anterior corneal radii of curvature, and 6.8 mm posterior corneal radii of curvature would have axial forward movement of 0.1 mm, 0.2 mm, and 0.3 mm corresponding to a myopic shift in refraction of

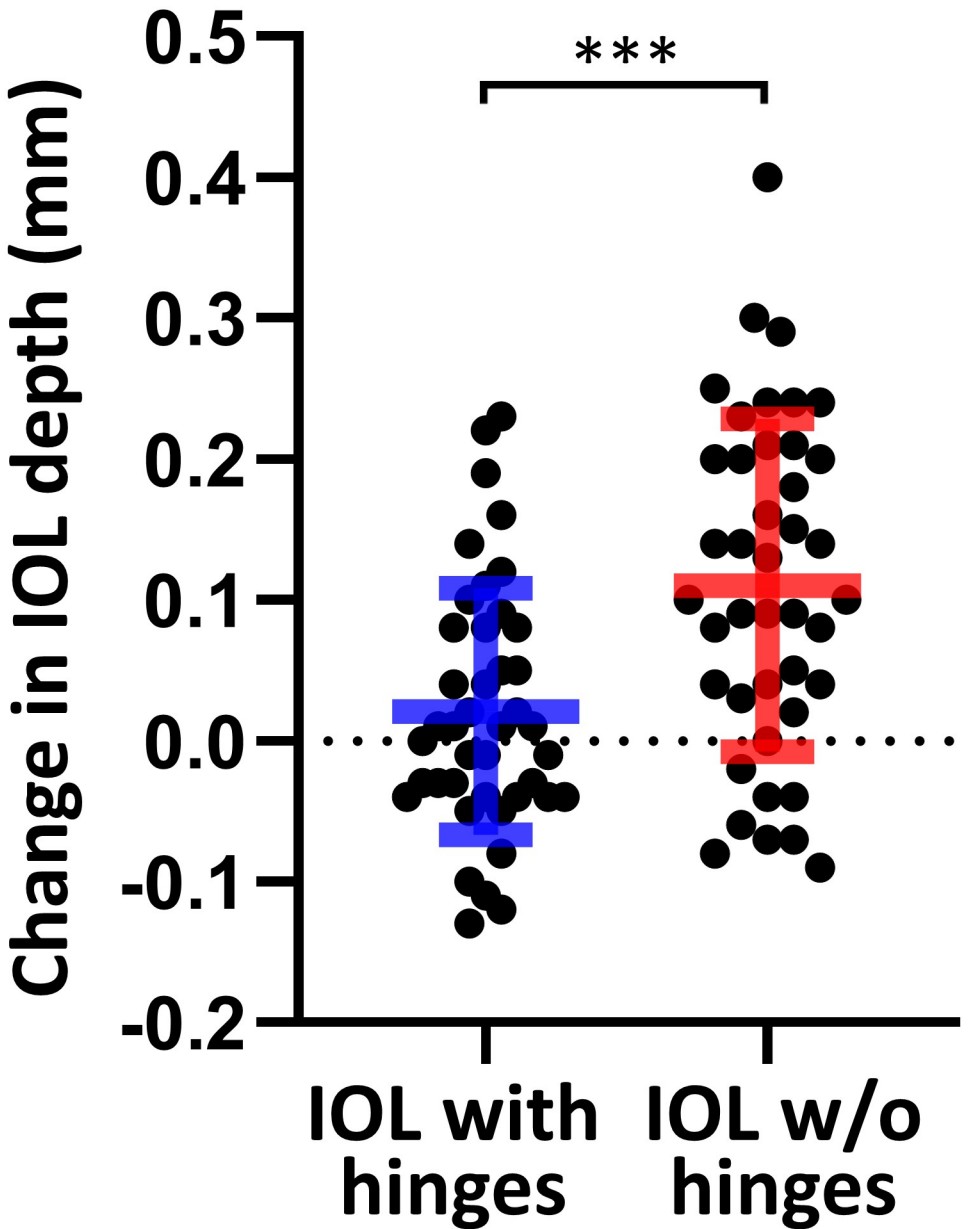

**Fig 3. Mean (± standard deviation) change in IOL depth between 4 days and 1 month after cataract surgery.** IOL, intraocular lens; W/O, without; Student *t*-test, ***$p < 0.001$.

0.13 D, 0.27 D, and 0.40 D, respectively [16]. The mean axial movement was less than 0.2 mm for both the IOL with hinges and IOL without hinges groups, which indicates a myopic shift of approximately 0.27 D based on modeling. Calculating the refractive change caused by the IOL shift in each subject demonstrates that the IOL shift strongly correlates with the changes in MRSE.

Klijn et al. evaluated the role of IOL position shift on long-term refractive shift from 1 month to 1 year after cataract surgery with implantation of Acrysof SA60AT IOL(Alcon) [17]. There was no correlation between the long-term change in refraction and the IOL position shift after cataract surgery, and Klijn et al. hypothesized that the postoperative refractive shift

**Table 3. Comparison of mean (± SD) anterior and posterior corneal power, change in each corneal power.**

| Parameters | IOL with hinges group n = 42 | IOL without hinges group n = 42 | P Value[a] |
|---|---|---|---|
| Anterior corneal power (D) | | | |
| 4 days postop. | 49.39 ± 1.69 | 48.84 ± 1.44 | 0.18 |
| 1 month postop. | 49.32 ± 1.70 | 48.86 ± 1.42 | 0.18 |
| P Value[b] | 0.67 | 0.48 | |
| Posterior corneal power (D) | | | |
| 4 days postop. | -6.40 ± 0.30 | -6.30 ± 0.27 | 0.10 |
| 1 month postop. | -6.34 ± 0.26 | -6.24 ± 0.23 | 0.10 |
| P Value[b] | 0.0003*** | 0.0015** | |
| Corneal real power (D) | | | |
| 4 days postop. | 43.04 ± 1.49 | 42.69 ± 1.27 | 0.25 |
| 1 month postop. | 43.11 ± 1.50 | 42.74 ± 1.25 | 0.23 |
| P Value[b] | 0.06 | 0.08 | |
| Change in anterior corneal power (D) | -0.02 ± 0.23 | -0.02 ± 0.20 | 0.89 |
| Change in posterior corneal power (D) | -0.07 ± 0.11 | -0.05 ± 0.10 | 0.55 |
| Change in corneal real power (D) | -0.07 ± 0.23 | -0.06 ± 0.21 | 0.79 |

*Statistically significant difference; P Value[a] between the 2 IOL groups; P Value[b] between the intervals.

might be explained by natural fluctuations in corneal curvature [17]. In contrast, the current results reveal a correlation between the short-term change in refraction and the IOL position shift from 4 days and 1 month after cataract surgery in the IOL without hinges group, but no correlation in the IOL with hinges group. Moreover, there was no significant correlation in corneal curvature with the short-term change in refraction of both groups. This suggests that there must be other factors that better explain short-term refractive changes after cataract surgery or multiple factors might be intricately interrelated.

In the current study, the changes in posterior corneal curvature were statistically significant but clinically insignificant (-0.07 ± 0.11 [IOL with hinges] and -0.05 ± 0.10 [IOL without hinges], Table 3), although posterior corneal curvature significantly flattened from day 4 to 1 month in both groups after surgery. This result is consistent with previous reports. Jin et al. reported that postoperative focal flattening in the posterior cornea were detected in the early postoperative period [18]. Although the change in postoperative corneal power is one of the factor related to the postoperative change in MRSE, there was no correlation between the myopic shift of MRSE and the change in corneal power during the early postoperative period in

**Table 4. Comparison of mean (± SD) change in manifest refractive spherical equivalent.**

| Parameters | IOL with hinges group n = 42 | IOL without hinges group n = 42 | P Value[a] |
|---|---|---|---|
| 4 days postop. | -1.08 ± 0.84 | -0.75 ± 0.83 | 0.08 |
| 1 month postop. | -1.19 ± 0.78 | -1.03 ± 0.79 | 0.34 |
| P Value[b] | 0.03* | < .0001*** | |
| Change in MRSE (D) | 0.11 ± 0.31 | 0.27 ± 0.36 | 0.03* |

MRSE = manifest refractive spherical equivalent;

*Statistically significant difference;

P Value[a] between the 2 IOL groups; P Value[b] between the intervals.

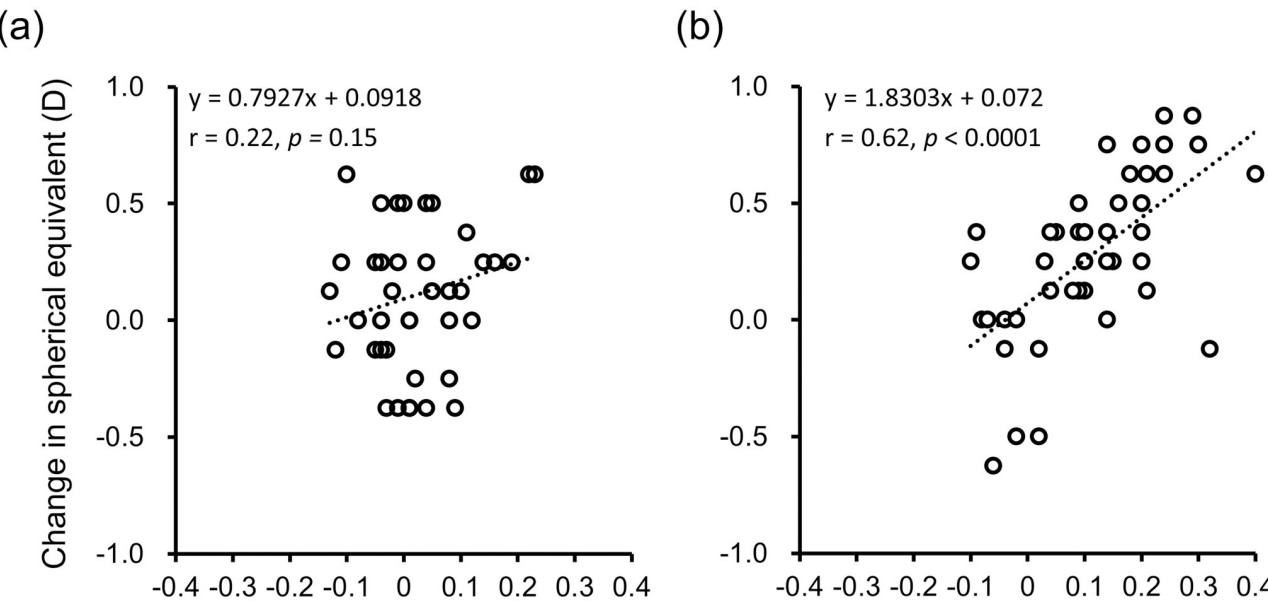

**Fig 4. Correlations with the changes in refraction and IOL depth between 4 days and 1 month after cataract surgery: (a) IOL with hinges and (b) IOL without hinges.**

the present study. Hence, the discrepancy between the refractive shift and corneal power change might be due other variables such as postoperative IOL shift. The postoperative stability of IOL position is one important factor for maintaining refractive stability.

This study has some limitations. First, postoperative ALs were not measured. Given the results of previous studies [9, 17, 19], it is unlikely that postoperative changes in AL are responsible for the changes in refraction. Second, two different IOL types were evaluated; namely, one was a mono-focal IOL and another was a toric mono-focal IOL. Future studies should be done with the same type of IOL. Third, the repeatability and reproducibility of AS-OCT measurements were not evaluated; however, this has been well proven in previous reports [20–22]. Future studies can also evaluate the effect of the change in IOL position and corneal value on the refractive prediction error of the IOL power calculation formula in a long-term study. Additionally, since subjective refraction is measured in 0.25 D steps, the results might vary with the examiner and patient. To compare postoperative changes in refraction, the use of 0.125 D steps for subjective refraction or an autorefractometer, which can measure values every 0.01 D, are options that can be used in the future [9].

In conclusion, despite similarities in material and planar haptics, postoperative IOL depth and axial movement can vary between two types of acrylic single-piece, open loop, foldable IOLs with different hinge designs at the early postoperative period (4 days to 1 month).

## Supporting information

**S1 Fig. Correlations between axial length and the changes in IOL depth between 4 days and 1 month after cataract surgery: (a) IOL with hinges and (b) IOL without hinges.**
(TIF)

**S1 File.**
(DOCX)

## Acknowledgments

The authors thank Keiko Ogawa, Saori Sugiyama, and Maki Matsumaru (Certified Orthoptists of National Hospital Organization, Tokyo Medical Center) for their assistance with data collection, and Joseph Wright and Heather S. Oliff, PhD for editorial assistance in the preparation of the manuscript.

## Author Contributions

**Conceptualization:** So Goto, Naoyuki Maeda, Kazuhiko Ohnuma, Toru Noda.

**Data curation:** So Goto.

**Formal analysis:** So Goto, Kazuhiko Ohnuma.

**Funding acquisition:** Toru Noda.

**Investigation:** So Goto, Kazuhiko Ohnuma.

**Methodology:** So Goto, Naoyuki Maeda, Toru Noda.

**Project administration:** So Goto.

**Resources:** So Goto.

**Software:** So Goto, Kazuhiko Ohnuma.

**Supervision:** Naoyuki Maeda, Kazuhiko Ohnuma, Toru Noda.

**Validation:** So Goto, Toru Noda.

**Visualization:** So Goto, Naoyuki Maeda, Toru Noda.

**Writing – original draft:** So Goto.

**Writing – review & editing:** So Goto, Naoyuki Maeda, Kazuhiko Ohnuma, Toru Noda.

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
