## [Decision Letter · Decision Letter 0]

6 Apr 2022

PONE-D-22-01418Comparison of two one-piece acrylic foldable intraocular lenses: Short-term change in axial movement after cataract surgery and its effect on refractionPLOS ONE

Dear Dr. Maeda,

Thank you for submitting your manuscript to PLOS ONE. After careful consideration, we feel that it has merit but does not fully meet PLOS ONE’s publication criteria as it currently stands. Therefore, we invite you to submit a revised version of the manuscript that addresses the points raised during the review process.

Please read the comments from Reviewer #2 and #3 carefully and consider their comments in the revised version.

We look forward to receiving your revised manuscript.

Kind regards,

Timo Eppig

Academic Editor

PLOS ONE

Journal Requirements:

“with funding from Alcon Japan Ltd (IIT # 54635107)”

We note that you have provided funding information within the Acknowledgements Section. Please note that funding information should not appear in the Acknowledgments section or other areas of your manuscript. We will only publish funding information present in the Funding Statement section of the online submission form.

 “The editorial assistance in the preparation of the manuscript was performed with funding from Alcon Japan Ltd (IIT # 54635107). The funders had no role in study design, data collection and analysis, decision to publish, or preparation of the manuscript.”

Reviewers' comments:

Reviewer's Responses to Questions

**Comments to the Author**

1. Is the manuscript technically sound, and do the data support the conclusions?

Reviewer #1: Yes

Reviewer #2: Yes

Reviewer #3: Yes

2. Has the statistical analysis been performed appropriately and rigorously? 

Reviewer #1: N/A

Reviewer #2: Yes

Reviewer #3: Yes

3. Have the authors made all data underlying the findings in their manuscript fully available?

Reviewer #1: No

Reviewer #2: Yes

Reviewer #3: Yes

4. Is the manuscript presented in an intelligible fashion and written in standard English?

Reviewer #1: Yes

Reviewer #2: Yes

Reviewer #3: Yes

5. Review Comments to the Author

Reviewer #1: 1. How did you assess you case number?

2. Was the CASIA measurement performed only once in each case?

3. Was it performed by one examiner? Or more?

Could you please provide ZEMAX simulation results or table as supplementary material?

Reviewer #2: Comparison of two one-piece acrylic foldable intraocular lenses: Short-term change in

axial movement after cataract surgery and its effect on refraction

The study describes axial IOL movement of two different IOLs from day 4 to 1 month postoperatively and correlates it with changes in the MRSE over time.

Methods:

When the study was retrospectively conducted, it is unlikely that it took 15 months. Rather patients which had surgery in this timespan were included.

Can you explain in short how the ZEMAX software works?

Did you use the post-OP Cataract module of the Casia 2 software?

Why did the authors include 42 eyes when only 34 were required according to their sample size calculation? Due to the retrospective nature of the study, no drop out correction is necessary.

Why did the authors assume 0.1mm as a SD? There are a lot of studies available with SD`s for axial movement of IOLs.

Since you study axial movement of the eye, you should clearly state which viscoelastics were used during the conduct of the study. Did the surgeon also remove the OVD from behind the lens? I ask this because one of the IOLs was a toric IOL, usually a surgeon removes the OVD behind the lens very thoroughly to avoid postoperative IOL rotation due to remaining viscoelastic.

Did the surgeon remove viscoelastics for both groups under the same manner? Or were the same viscoealstics used? This could affect postoperative axial stabilization within the first weeks.

How was the postoperative refraction (MRSE) assessed at 4 days and 1 month? Objective or subjective refraction?

I miss a detailed description of both IOLs in the methods section (overall diameter, optic diameter, capsular bag angle, haptic resilience, and other characteristics)

Results:

It’s a little bit confusing that you present change in IOL depth and actual axial movement. What`s the difference between these two variables? You should describe this in detail in the methods.

Are the results absolute values? Since the IOL can move forward and backwards? This is a very essential point

I would also report the range for the IOL depths and/or IOL axial movements [min;max] or seeing Boxplots of the results to get a better feeling were all the results are located.

Discussion:

What do the authors mean by “constricted” and “straight” haptics, what exactly is the difference between the haptics of these two IOLs? A figure would be good for better understandability.

The overall discussion is well written.

I miss a single point: Since both of the IOLs show pretty much the same design, what may be the main reason or multiple reasons for this difference between the IOLs in axial movement?

Another important point: was IOL movement correlated with axial length? It has been shown in earlier studies, that shorter eyes (or shorter preoperative ACDs) are more prone to axial IOL movement postoperatively. (Schartmüller et al, JRS 2021) (Schartmüller et al ESCRS 2019 Paris)

Reviewer #3: The authors present an interesting manuscript about two different IOL (haptic) designs and their axial stability in the postoperative course. Results are set in context with refractive changes. Interestingly, the authors chose a toric lens and a nontoric lens, which leads to differences in the centration of the haptics, still I like the study design.

While I think that overall, the manuscript is very well composed, I do see a few open questions:

With Aquaeous depth as main criterion, the authors reported significant changes in posterior radial curvature but failed to report changes in CCT and in ACD. Changes in posterior corneal curvature might be a sign of swelling of the corneal endothelium, certain levels of Descemet striae and swelling are not seldom after cataract surgery. AQD should not be described without ACD and CCT.

What is lacking is a discussion about the ideal timeframe for lens constant optimization, as a shift of the IOL and refraction obviously influence constant optimization if refractive results for certain types of IOLs are analyzed too early.

I would think that an autorefractometer is influenced by the Abbe number of the IOL and a diffractive form. We see systematically differing results from subjective refraction that differ for various IOLs. I don’t necessarily advise to change to ARF. Furthermore, repeatability has proven better for us for subjective refraction than for pseudophakic objective refraction. It is of note that we actually have 0.125 Dpt steps for subjective refraction (although not for a phoropter) that can be used if 0.25 Dpt seems to be too imprecise.

---

## [Author Response · Author response to Decision Letter 0]

20 May 2022

All reviewers:

First of all, I would appreciate it if you could see the submitted response file. Just in case, the following is a transcription of the response without the figures.

Previously each IOL evaluated in this study was described by the product name. Based on the reviewers’ point, we are now referencing the products with descriptors “with hinges or without hinges” in lieu of product names to better clarify the argument of this manuscript.

Reviewer #1:

1. How did you assess you case number?

Response: We performed the sample size power calculation. At the end of the method section, we mentioned as follows; The sample size was calculated to detect a difference in error of 0.10 mm between 2 groups; with a signiﬁcance level of 5%, a statistical power of 80%, and assuming standard deviation (SD) to 0.1 mm, 34 eyes were required. Page 7, line 132.

2. Was the CASIA measurement performed only once in each case?

Response: Thank you for this comment. The CASIA was performed three times in each visit and data is exported as the mean value of three measurements. We have added the explanation in the Methods section. Page6, line 111.

3. Was it performed by one examiner? Or more?

Response: Thank you for this comment. Yes, the CASIA was performed by one experienced ophthalmic technician. We have added clarification to the Methods section. Page 6, line 111.

4. Could you please provide ZEMAX simulation results or table as supplementary material?

Response: Thank you for this comment. In Table 2, we have already described the simulation results of refraction change associated with change in IOL position, those were estimated using OpticStudio 16.5 Sp5 (Zemax, LLC.), the industry standard optical system design software; however, the representation of the parameter was not clear. Therefore, we have changed “Estimated refractive error associated with change in IOL depth” to “Refractive change associated with change in IOL depth” and have added the phrase of “C calculated by the OpticStudio software” in the footnote of Table 2.

Reviewer #2:

Methods: When the study was retrospectively conducted, it is unlikely that it took 15 months. Rather patients which had surgery in this timespan were included.

Response: Thank you for this comment. As the reviewer pointed out, this was a retrospective, observational, comparative, single-center study of all patients who had undergone uncomplicated cataract surgery at the National Hospital Organization, Tokyo Medical Center, Tokyo, Japan, between December 2015 and February 2017. We corrected the sentence at the beginning of the Methods section. Page 5, line 74.

Can you explain in short how the ZEMAX software works?

Response: The Zemax software, the official name is "OpticStudio" [Zemax, LLC.], is the premier optical design software used by engineers around the globe for 30 years. We have corrected the name of the software in the manuscript. We can estimate the theoretical postoperative refractive change that is caused by postoperative axial movement of IOL by using the OpticStudio software (see the attached screenshot of OpticStudio software). For example, if one eye showed 0.2 mm forward shift of IOL between postoperative 4 days and 1 month, the refractive changes caused by the change in IOL position are affected by the IOL power, corneal power, and axial length. Eventually, we can calculate the estimated refractive change associated with change in IOL depth. The results of these simulations using the OpticStudio was shown in Table 2 as “Refractive change associated with change in IOL depth (D)c.” “C” is explained in the footnote of Table 2.

Did you use the post-OP Cataract module of the Casia 2 software?

Response: Yes, we used the post-op cataract module of the Casia2, which gave us the postoperative aqueous depth, corneal thickness, and anterior chamber depth as well.

Why did the authors include 42 eyes when only 34 were required according to their sample size calculation? Due to the retrospective nature of the study, no drop out correction is necessary.

Response: Thank you for this comment. Before the current study started, we had a data set of 42 eyes for each group. When we proposed this study, we performed a sample size calculation to confirm whether the subject number was enough for the investigation of the topic.

In this revision, we have randomly selected 34 eyes from 42 eyes, and analyzed the data. We have validated this analysis three times. As the result, every data set demonstrated significant difference shown in the table below.

Comparison of IOL depth between IQ Toric and XY-1

n = 34 IQ Toric XY-1 P value

1st 0.014 ± 0.076 0.098 ± 0.127 0.0015

2nd 0.023 ± 0.090 0.096 ± 0.126 0.0077

3rd 0.029 ± 0.018 0.129 ± 0.018 0.0002

34 eyes were randomly selected from 42 eyes

In addition, as we mentioned the detail in the next question below, once we use the SD of 0.11, the power calculation tells us that 41 subjects are needed. Based on these calculation result, the data set of 42 subjects seems to be appropriate for this investigation.

Why did the authors assume 0.1mm as a SD? There are a lot of studies available with SD`s for axial movement of IOLs.

Response: Thank you for this constructive comment. To determine the SD of the change in IOL position to perform the power calculation, we referred to the article published by Hayashi K, et al. (doi:10.1016/j.ajo.2020.05.031.). Given that the SD of the change in IOL position in the paper was 0.11mm, we should have use 0.11 mm for SD. If we performed sample size power analysis to detect a difference in error of 0.1 mm between two groups; with a signiﬁcance level of 5%, a statistical power of 80%, and assuming standard deviation (SD) to 0.11 mm, 41 eyes were required. Based on these data, we concluded that 42 subjects are appropriate to investigate the postoperative change in IOL depth. We have edited the sentence related to power sample analysis at the end of the Methods section. Page 6, line 132.

Since you study axial movement of the eye, you should clearly state which viscoelastics were used during the conduct of the study. Did the surgeon also remove the OVD from behind the lens? I ask this because one of the IOLs was a toric IOL, usually a surgeon removes the OVD behind the lens very thoroughly to avoid postoperative IOL rotation due to remaining viscoelastic.

Did the surgeon remove viscoelastics for both groups under the same manner? Or were the same viscoealstics used? This could affect postoperative axial stabilization within the first weeks.

Response: Thank you for this insightful comment. I have added the information of OVD. Additionally, the surgeon removed the same OVD with the same manner for both groups. Briefly, the I/A tip is inserted behind the IOL optic and the posterior chamber is directly irrigated and cleaned. I have added the explanations of these procedures in the Methods section. Page 5, line 94.

How was the postoperative refraction (MRSE) assessed at 4 days and 1 month? Objective or subjective refraction?

Response: Thank you for this comment. Subjective MRSE was assessed at 4 days and 1 month.

I miss a detailed description of both IOLs in the methods section (overall diameter, optic diameter, capsular bag angle, haptic resilience, and other characteristics)

Response: Thank you for this comment. We have added the information of both IOLs in the Methods section of the manuscript with the photograph of both IOLs. Both of the IOLs made of acrylic material have 13.0 mm overall diameter, 6.0 mm optic diameter, and planar-haptic design, whereas IQ Toric IOL has a specific flexible hinge design. We have added the information of each IOL in the Methods section (page 5, line 92). There is no available information of haptic resilience.

Results:

It’s a little bit confusing that you present change in IOL depth and actual axial movement. What`s the difference between these two variables? You should describe this in detail in the methods. Are the results absolute values? Since the IOL can move forward and backwards? This is a very essential point.

Response: Thank you for this constructive comment. Whereas the change in IOL depth is the numerical value, the actual axial movement is the absolute value. As the reviewer pointed out the definitions of these parameters are a little confusing, we have replaced “actual axial movement” with “absolute axial movement” in Table 2.

I would also report the range for the IOL depths and/or IOL axial movements [min;max] or seeing Boxplots of the results to get a better feeling were all the results are located.

Response: Thank you for this constructive comment. I have added the data of [min:max] in Table 2 and have made the graph including all dots as Figure 3 (see below).

Fig 3. Fig 3. Mean (± standard deviation) change in IOL depth between 4 days and 1 month after cataract surgery. IOL, intraocular lens; W/O, without; Student t-test, ***p < 0.001

Discussion:

What do the authors mean by “constricted” and “straight” haptics, what exactly is the difference between the haptics of these two IOLs? A figure would be good for better understandability.

Response: We appreciate this valuable comment. We have added the photographs of IOLs as Fig 1 that emphasizes the difference of the haptics structure.

Fig 1. Optic and haptic configurations of (a) IOL with hinges and (b) IOL without hinges. While both of IOLs have planar haptics designs with a 0-degree angle, the haptics of the IOL with hinges (c) are constricted (i.e. flexible hinge design, black arrows) and the haptics of the IOL without hinges (d) are straight.

The overall discussion is well written. I miss a single point: Since both of the IOLs show pretty much the same design, what may be the main reason or multiple reasons for this difference between the IOLs in axial movement?

Response: Thank you for this constructive comment. Both of IOLs have planar-haptic design, whereas IQ Toric IOL has a specific flexible hinge design (Fig 1. C, black arrows). Based on an in vitro experiment (IOL compression test), the planar haptics with a flexible hinge design could minimize axial forces and allow the IOLs to remain planar when compressed. 13 Although the axial stability of IOLs has been proved in vitro, there was less clinical data. We believe that the difference in haptic configurations would be the main factor that provides the postoperative stability.

13. Lane S, Collins S, Das KK, et al. Evaluation of intraocular lens mechanical stability. J Cataract Refract Surg 2019;45:501–506.

Another important point: was IOL movement correlated with axial length? It has been shown in earlier studies, that shorter eyes (or shorter preoperative ACDs) are more prone to axial IOL movement postoperatively. (Schartmüller et al, JRS 2021) (Schartmüller et al ESCRS 2019 Paris)

Response: Thank you for this constructive comment. We have confirmed the relationship between axial length and axial IOL movement, indicating that longer eyes are more prone to axial IOL movement postoperatively in only XY-1 (IOL without hinges) group, which is the opposite result of Schartmüller et al, JRS 2021. There are several differences between our study and Schartmüller et al, such as the observation period and the type of IOL. We assumed that longer eye tends to have larger bag, and as such IOLs can more easily to move in the bag postoperatively. At least, the result shown as S1 Fig might also indicate that IQ Toric IOLs (IOL with hinges) are more stable. We have added this result to the Discussion section and added supplemental figure 1. Page12, line 194.

S1 Fig. Correlations between axial length and the changes in IOL depth between 4 days and 1 month after cataract surgery: (a) IOL with hinges and (b) IOL without hinges.

Reviewer #3:

The authors present an interesting manuscript about two different IOL (haptic) designs and their axial stability in the postoperative course. Results are set in context with refractive changes. Interestingly, the authors chose a toric lens and a nontoric lens, which leads to differences in the centration of the haptics, still I like the study design.

While I think that overall, the manuscript is very well composed, I do see a few open questions: With Aqueous depth as main criterion, the authors reported significant changes in posterior radial curvature but failed to report changes in CCT and in ACD. Changes in posterior corneal curvature might be a sign of swelling of the corneal endothelium, certain levels of Descemet striae and swelling are not seldom after cataract surgery. AQD should not be described without ACD and CCT.

Response: Thank you for providing this insightful comment. As the reviewer recommended, we have analyzed the changes in CCT and ACD. Whereas there was no significant difference in change in CCT between 2 groups (p = 0.31), the change in ACD was significantly smaller in IQ Toric (IOL with hinges) group than XY-1 (IOL without hinges) group, which is the same as the result of IOL depth (i.e. postoperative AQD). We have added these results in Table 2. Subsequently, the change in ACD was significantly correlated with the refractive changes in XY-1 group (r = 0.63, p < 0.0001), but there was no significant difference in IQ Toric group (r = 0.21, p = 0.20); this result is also the same as the change in AQD.

What is lacking is a discussion about the ideal timeframe for lens constant optimization, as a shift of the IOL and refraction obviously influence constant optimization if refractive results for certain types of IOLs are analyzed too early.

Response: Thank you for this valuable comment. We agree with the reviewer’s comment. Therefore, we have added the following discussion in the Discussion section: The postoperative stability of IOL position is one important factor for maintaining refractive stability. Page 13, line 230.

The purpose of the current study is to investigate the postoperative IOL axial movement, not improve the prediction error of the IOL power calculation formulas. Even though there are a lot of IOL formulas, we can say that at least 1 month or more after surgery is required to evaluate the postoperative refraction as many studies reported.

I would think that an autorefractometer is influenced by the Abbe number of the IOL and a diffractive form. We see systematically differing results from subjective refraction that differ for various IOLs. I don’t necessarily advise to change to ARF. Furthermore, repeatability has proven better for us for subjective refraction than for pseudophakic objective refraction. It is of note that we actually have 0.125 Dpt steps for subjective refraction (although not for a phoropter) that can be used if 0.25 Dpt seems to be too imprecise.

Response: We truly appreciate your comments. As you recommended, we would like to have 0.125 D steps for subjective refraction, which would give us more promising results in this field. I have added this recommendation in the Discussion section. Page14, line 241.

---

## [Decision Letter · Decision Letter 1]

7 Jul 2022

PONE-D-22-01418R1Comparison of two one-piece acrylic foldable intraocular lenses: Short-term change in axial movement after cataract surgery and its effect on refractionPLOS ONE

Dear Dr. Maeda,

Thank you for submitting your manuscript to PLOS ONE. After careful consideration, we feel that it has merit but does not fully meet PLOS ONE’s publication criteria as it currently stands. Therefore, we invite you to submit a revised version of the manuscript that addresses the points raised during the review process.

Please carefully read the comments from reviewer #3 and consider these comments in the revised version. Especially the comments regarding the model eye and the use of OpticStudio should be addressed.

We look forward to receiving your revised manuscript.

Kind regards,

Timo Eppig

Academic Editor

PLOS ONE

Journal Requirements:

Reviewers' comments:

Reviewer's Responses to Questions

**Comments to the Author**

1. If the authors have adequately addressed your comments raised in a previous round of review and you feel that this manuscript is now acceptable for publication, you may indicate that here to bypass the “Comments to the Author” section, enter your conflict of interest statement in the “Confidential to Editor” section, and submit your "Accept" recommendation.

Reviewer #1: All comments have been addressed

Reviewer #3: (No Response)

2. Is the manuscript technically sound, and do the data support the conclusions?

Reviewer #1: Partly

Reviewer #3: Yes

3. Has the statistical analysis been performed appropriately and rigorously? 

Reviewer #1: I Don't Know

Reviewer #3: Yes

4. Have the authors made all data underlying the findings in their manuscript fully available?

Reviewer #1: Yes

Reviewer #3: Yes

5. Is the manuscript presented in an intelligible fashion and written in standard English?

Reviewer #1: Yes

Reviewer #3: Yes

6. Review Comments to the Author

Reviewer #1: The authors have properly addressed all comments. The manuscript is now improved in all its part. I suggest to send the exact ZEMAX table as Supplementary materials.

Reviewer #3: The manuscript looks promising in its current state. Here are my last comments after the revision:

Reviewer comments:

o I didn’t view this as critical in my first review, but as both other reviewers noticed this point I have to agree that the description of how OpticStudio was used is kind of lacking in this study. Apparently the authors used a simulation model with 7 rays? Which model eye was chosen? Did the authors use full aperture raytracing or the paraxial simplification form? Was the individual pupil size and asphericity considered? Considering 2 different IOL designs were used was IOL geometry data known?

At least all the actual simulation details used for this analysis should be provided in the materials section.

o Regarding the answer to my own comments of IOL constants optimization:

“The purpose of the current study is to investigate the postoperative IOL axial movement, not improve the prediction error of the IOL power calculation formulas.”

Well, why else are we interested in axial IOL movement, if not for right (right lens constant) and stable (good IOL design) refractive results?

Anyways, “we can say that at least 1 month or more after surgery is required to evaluate the postoperative refraction as many studies reported” is a sufficient statement, this doesn’t have to go into an in-depth analysis of every IOL calculation formula.

Methods:

o I think the statement “the industry standard optical system design” sounds a bit like advertisement, can this be omitted?

o I figure the refraction lane length was standardized? Which lane length was used?

Otherwise, this study can be accepted in my eyes.

7. PLOS authors have the option to publish the peer review history of their article (what does this mean?). If published, this will include your full peer review and any attached files.

Reviewer #1: No

Reviewer #3: No

---

## [Author Response · Author response to Decision Letter 1]

1 Aug 2022

Manuscript ID: PONE-D-22-01418

Title: Comparison of two one-piece acrylic foldable intraocular lenses: Short-term change in axial movement after cataract surgery and its effect on refraction

Point-by-Point Response 2

Reviewer #1:

The authors have properly addressed all comments. The manuscript is now improved in all its part. I suggest to send the exact ZEMAX table as Supplementary materials.

Response: Thank you for this constructive comment. At the reviewer’s suggestion, we have added supporting information, which includes an exact ZEMAX table based on the “zmx” file and screenshots of the Lens Data Editor and Effective Focal Length (EFL) and axial Longitudinal Spherical Aberration (LSA0). Page 6, line 121.

Supporting Information 

Method:

Zemax OpticStudio (ZOS) was used to evaluate the refractive errors from the change in postoperative lens position (the difference between 4 days postop and 1 month postop).

All ocular biometric parameters were obtained from preoperative measurement data and assumed to be constant in order to not interfere with the lens position variability.

The procedures were as follows:

1. Cornea anterior radius curvature was calculated by 337.5/PreOp(AveK).

2. Cornea posterior radius curvature was assumed to be constant for all subjects, and defined as 6.5 mm.

3. Cornea center thickness was assumed to be constant for all subjects, and defined as 550 mm.

4. Cornea and aqueous refractive indices were defined as 1.376 and 1.336, respectively. 

5. IOL design and refractive indices were given by the manufacturers.

6. Axial power was obtained by PreOp(AL).

7. Lens positions were determined using the measurement results at 4 days and 1 month.

8. Back focal length (the distance from the posterior IOL to the retina) was then calculated using the aforementioned distances.

All the information was inputted into the ZOS Lens Data Editor, and ZOS calculated the Effective Focal Length (EFL) and axial Longitudinal Spherical Aberration (LSA0) (Supplemental figure).

The refractive error was calculated using the following formula

SE = -(1000/EFL – 1000/(EFL-LSA0))

 

Raw data of ZOSModel.zmx:

MODE SEQ

NAME 

PFIL 0 0 0

LANG 0

UNIT MM X W X CM MR CPMM

FLOA

ENVD 35 1 1

GFAC 0 0

GCAT SCHOTT 

RAIM 0 2 1 1 0 0 0 0 0 1

PUSH -2.7131892210931558e-05 0 0 0 0 0

SDMA 0 1 0

OMMA 1 1

FTYP 0 0 1 1 0 0 0

ROPD 2

HYPR 0

PICB 1

XFLN 0 0 0 0 0 0 0 0 0 0 0 0

YFLN 0 0 0 0 0 0 0 0 0 0 0 0

FWGN 1 1 1 1 1 1 1 1 1 1 1 1

VDXN 0 0 0 0 0 0 0 0 0 0 0 0

VDYN 0 0 0 0 0 0 0 0 0 0 0 0

VCXN 0 0 0 0 0 0 0 0 0 0 0 0

VCYN 0 0 0 0 0 0 0 0 0 0 0 0

VANN 0 0 0 0 0 0 0 0 0 0 0 0

WAVM 1 0.54607399999999995 1

WAVM 2 0.55000000000000004 1

WAVM 3 0.55000000000000004 1

WAVM 4 0.55000000000000004 1

WAVM 5 0.55000000000000004 1

WAVM 6 0.55000000000000004 1

WAVM 7 0.55000000000000004 1

WAVM 8 0.55000000000000004 1

WAVM 9 0.55000000000000004 1

WAVM 10 0.55000000000000004 1

WAVM 11 0.55000000000000004 1

WAVM 12 0.55000000000000004 1

WAVM 13 0.55000000000000004 1

WAVM 14 0.55000000000000004 1

WAVM 15 0.55000000000000004 1

WAVM 16 0.55000000000000004 1

WAVM 17 0.55000000000000004 1

WAVM 18 0.55000000000000004 1

WAVM 19 0.55000000000000004 1

WAVM 20 0.55000000000000004 1

WAVM 21 0.55000000000000004 1

WAVM 22 0.55000000000000004 1

WAVM 23 0.55000000000000004 1

WAVM 24 0.55000000000000004 1

PWAV 1

POLS 1 0 1 0 0 1 0

GLRS 3 0

GSTD 0 100.000 100.000 100.000 100.000 100.000 100.000 0 1 1 0 0 1 1 1 1 1 1

NSCD 100 500 0 0.001 5 9.9999999999999995e-07 0 0 0 0 0 0 1000000 0 2

COFN QF "COATING.DAT" "SCATTER_PROFILE.DAT" "ABG_DATA.DAT" "PROFILE.GRD"

COFN COATING.DAT SCATTER_PROFILE.DAT ABG_DATA.DAT PROFILE.GRD

SURF 0

 COMM Object Distance

 TYPE STANDARD

 FIMP 

 CURV 0.0 0 0 0 0 ""

 HIDE 0 0 0 0 0 0 0 0 0 0

 MIRR 2 0

 SLAB 9

 DISZ INFINITY

 DIAM 0 0 0 0 1 ""

 MEMA 0 0 0 0 1 ""

 POPS 0 0 0 0 0 0 0 0 1 1 1 1 0 0 0 0

SURF 1

 COMM Cornea Anterior

 TYPE STANDARD

 FIMP 

 CURV 1.386962962962962342E-01 0 0 0 0 ""

 HIDE 0 0 0 0 0 0 0 0 0 0

 MIRR 2 0

 SLAB 1

 DISZ 0.55000000000000004

 GLAS ___BLANK 1 0 1.3759999999999999 0 0 0 0 0 0 0 

 DIAM 5 1 0 0 1 ""

 MEMA 5 0 0 0 1 ""

 POPS 0 0 0 0 0 0 0 0 1 1 1 1 0 0 0 0

 FLAP 0 5 0

SURF 2

 COMM Cornea Posterior

 TYPE STANDARD

 FIMP 

 CURV 1.538461538461538547E-01 0 0 0 0 ""

 HIDE 0 1 0 0 0 0 0 0 0 0

 MIRR 2 0

 SLAB 2

 DISZ 3.8639999999999999

 GLAS ___BLANK 1 0 1.3360000000000001 0 0 0 0 0 0 0 

 DIAM 5 1 0 0 1 ""

 MEMA 5 0 0 0 1 ""

 POPS 0 0 0 0 0 0 0 0 1 1 1 1 0 0 0 0

 FLAP 0 5 0

SURF 3

 COMM Pupil

 STOP

 TYPE STANDARD

 FIMP 

 CURV 0.0 0 0 0 0 ""

 HIDE 0 1 0 0 0 0 0 0 0 0

 MIRR 2 0

 SLAB 3

 DISZ 0

 GLAS ___BLANK 2 2 1.3360000000000001 0 0 0 0 0 0 0 

 DIAM 1.5 1 0 0 1 ""

 MEMA 1.5 1 0 0 1 ""

 POPS 0 0 0 0 0 0 0 0 1 1 1 1 0 0 0 0

 FLAP 0 1.5 0

SURF 4

 COMM IOL Anterior

 TYPE STANDARD

 FIMP 

 CURV 5.263157894736841813E-02 0 0 0 0 ""

 HIDE 0 0 0 0 0 0 0 0 0 0

 MIRR 2 0

 SLAB 4

 DISZ 0.69999999999999996

 GLAS ___BLANK 1 0 1.5 0 0 0 0 0 0 0 

 DIAM 3 1 0 0 1 ""

 MEMA 3 1 0 0 1 ""

 POPS 0 0 0 0 0 0 0 0 1 1 1 1 0 0 0 0

 FLAP 0 3 0

SURF 5

 COMM IOL Posterior

 TYPE STANDARD

 FIMP 

 CURV -5.263157894736841813E-02 0 0 0 0 ""

 HIDE 0 1 0 0 0 0 0 0 0 0

 MIRR 2 0

 SLAB 5

 DISZ 17.274000000000001

 GLAS ___BLANK 2 2 1.3360000000000001 0 0 0 0 0 0 0 

 DIAM 3 1 0 0 1 ""

 MEMA 3 1 0 0 1 ""

 POPS 0 0 0 0 0 0 0 0 1 1 1 1 0 0 0 0

 FLAP 0 3 0

SURF 6

 TYPE STANDARD

 FIMP 

 CURV 0.0 0 0 0 0 ""

 HIDE 0 0 0 0 0 0 0 0 0 0

 MIRR 2 0

 SLAB 8

 DISZ 0

 DIAM 0.040796963281374721 0 0 0 1 ""

 MEMA 5 0 0 0 1 ""

 POPS 0 0 0 0 0 0 0 0 1 1 1 1 0 0 0 0

EFFL 0 1 0 0 0 0 0 0 0 0

LONA 0 0 0 0 0 0 0 0 0 0

TOL TOFF 0 0 0 0 0 0 0 0

MNUM 1 1

MOFF 0 1 "" 0 0 0 1 1 0 0.0 "" 0

 

Reviewer #3:

I didn’t view this as critical in my first review, but as both other reviewers noticed this point I have to agree that the description of how OpticStudio was used is kind of lacking in this study. Apparently the authors used a simulation model with 7 rays? Which model eye was chosen? Did the authors use full aperture raytracing or the paraxial simplification form? Was the individual pupil size and asphericity considered? Considering 2 different IOL designs were used was IOL geometry data known? At least all the actual simulation details used for this analysis should be provided in the materials section.

Response:

Thank you for this constructive comment. As the reviewer pointed out, we have added the details related to how to analyze the data using ZEMAX software in the Method section as follows (page 6, line 115):

The calculations with OpticStudio were performed in the paraxial form. Therefore, the number of rays and aperture size did not contribute to the calculated results. Although asphericity affects the spherical results when comparing 4 days and 1 month spherical differences, to simplify the analysis, asphericity can be removed from the calculation of spherical differences. The eye model was built from the patients' biometry data including the cornea power, aqueous depth, and AL (refer to the Supporting Information).

Additionally, we have added the following “Supporting Information” file to describe the details of how to analyze the data with the OpticStudio.

Supporting Information

Method:

Zemax OpticStudio (ZOS) was used to evaluate the refractive errors from the change in postoperative lens position (the difference between 4 days postop and 1 month postop).

All ocular biometric parameters were obtained from preoperative measurement data and assumed to be constant in order to not interfere with the lens position variability.

The procedures were as follows:

9. Cornea anterior radius curvature was calculated by 337.5/PreOp(AveK).

10. Cornea posterior radius curvature was assumed to be constant for all subjects, and defined as 6.5 mm.

11. Cornea center thickness was assumed to be constant for all subjects, and defined as 550 mm.

12. Cornea and aqueous refractive indices were defined as 1.376 and 1.336, respectively. 

13. IOL design and refractive indices were given by the manufacturers.

14. Axial power was obtained by PreOp(AL).

15. Lens positions were determined using the measurement results at 4 days and 1 month.

16. Back focal length (the distance from the posterior IOL to the retina) was then calculated using the aforementioned distances.

All the information was inputted into the ZOS Lens Data Editor, and ZOS calculated the Effective Focal Length (EFL) and axial Longitudinal Spherical Aberration (LSA0) (Supplemental figure).

The refractive error was calculated using the following formula

SE = -(1000/EFL – 1000/(EFL-LSA0))

 

Raw data of ZOSModel.zmx:

MODE SEQ

NAME 

PFIL 0 0 0

LANG 0

UNIT MM X W X CM MR CPMM

FLOA

ENVD 35 1 1

GFAC 0 0

GCAT SCHOTT 

RAIM 0 2 1 1 0 0 0 0 0 1

PUSH -2.7131892210931558e-05 0 0 0 0 0

SDMA 0 1 0

OMMA 1 1

FTYP 0 0 1 1 0 0 0

ROPD 2

HYPR 0

PICB 1

XFLN 0 0 0 0 0 0 0 0 0 0 0 0

YFLN 0 0 0 0 0 0 0 0 0 0 0 0

FWGN 1 1 1 1 1 1 1 1 1 1 1 1

VDXN 0 0 0 0 0 0 0 0 0 0 0 0

VDYN 0 0 0 0 0 0 0 0 0 0 0 0

VCXN 0 0 0 0 0 0 0 0 0 0 0 0

VCYN 0 0 0 0 0 0 0 0 0 0 0 0

VANN 0 0 0 0 0 0 0 0 0 0 0 0

WAVM 1 0.54607399999999995 1

WAVM 2 0.55000000000000004 1

WAVM 3 0.55000000000000004 1

WAVM 4 0.55000000000000004 1

WAVM 5 0.55000000000000004 1

WAVM 6 0.55000000000000004 1

WAVM 7 0.55000000000000004 1

WAVM 8 0.55000000000000004 1

WAVM 9 0.55000000000000004 1

WAVM 10 0.55000000000000004 1

WAVM 11 0.55000000000000004 1

WAVM 12 0.55000000000000004 1

WAVM 13 0.55000000000000004 1

WAVM 14 0.55000000000000004 1

WAVM 15 0.55000000000000004 1

WAVM 16 0.55000000000000004 1

WAVM 17 0.55000000000000004 1

WAVM 18 0.55000000000000004 1

WAVM 19 0.55000000000000004 1

WAVM 20 0.55000000000000004 1

WAVM 21 0.55000000000000004 1

WAVM 22 0.55000000000000004 1

WAVM 23 0.55000000000000004 1

WAVM 24 0.55000000000000004 1

PWAV 1

POLS 1 0 1 0 0 1 0

GLRS 3 0

GSTD 0 100.000 100.000 100.000 100.000 100.000 100.000 0 1 1 0 0 1 1 1 1 1 1

NSCD 100 500 0 0.001 5 9.9999999999999995e-07 0 0 0 0 0 0 1000000 0 2

COFN QF "COATING.DAT" "SCATTER_PROFILE.DAT" "ABG_DATA.DAT" "PROFILE.GRD"

COFN COATING.DAT SCATTER_PROFILE.DAT ABG_DATA.DAT PROFILE.GRD

SURF 0

 COMM Object Distance

 TYPE STANDARD

 FIMP 

 CURV 0.0 0 0 0 0 ""

 HIDE 0 0 0 0 0 0 0 0 0 0

 MIRR 2 0

 SLAB 9

 DISZ INFINITY

 DIAM 0 0 0 0 1 ""

 MEMA 0 0 0 0 1 ""

 POPS 0 0 0 0 0 0 0 0 1 1 1 1 0 0 0 0

SURF 1

 COMM Cornea Anterior

 TYPE STANDARD

 FIMP 

 CURV 1.386962962962962342E-01 0 0 0 0 ""

 HIDE 0 0 0 0 0 0 0 0 0 0

 MIRR 2 0

 SLAB 1

 DISZ 0.55000000000000004

 GLAS ___BLANK 1 0 1.3759999999999999 0 0 0 0 0 0 0 

 DIAM 5 1 0 0 1 ""

 MEMA 5 0 0 0 1 ""

 POPS 0 0 0 0 0 0 0 0 1 1 1 1 0 0 0 0

 FLAP 0 5 0

SURF 2

 COMM Cornea Posterior

 TYPE STANDARD

 FIMP 

 CURV 1.538461538461538547E-01 0 0 0 0 ""

 HIDE 0 1 0 0 0 0 0 0 0 0

 MIRR 2 0

 SLAB 2

 DISZ 3.8639999999999999

 GLAS ___BLANK 1 0 1.3360000000000001 0 0 0 0 0 0 0 

 DIAM 5 1 0 0 1 ""

 MEMA 5 0 0 0 1 ""

 POPS 0 0 0 0 0 0 0 0 1 1 1 1 0 0 0 0

 FLAP 0 5 0

SURF 3

 COMM Pupil

 STOP

 TYPE STANDARD

 FIMP 

 CURV 0.0 0 0 0 0 ""

 HIDE 0 1 0 0 0 0 0 0 0 0

 MIRR 2 0

 SLAB 3

 DISZ 0

 GLAS ___BLANK 2 2 1.3360000000000001 0 0 0 0 0 0 0 

 DIAM 1.5 1 0 0 1 ""

 MEMA 1.5 1 0 0 1 ""

 POPS 0 0 0 0 0 0 0 0 1 1 1 1 0 0 0 0

 FLAP 0 1.5 0

SURF 4

 COMM IOL Anterior

 TYPE STANDARD

 FIMP 

 CURV 5.263157894736841813E-02 0 0 0 0 ""

 HIDE 0 0 0 0 0 0 0 0 0 0

 MIRR 2 0

 SLAB 4

 DISZ 0.69999999999999996

 GLAS ___BLANK 1 0 1.5 0 0 0 0 0 0 0 

 DIAM 3 1 0 0 1 ""

 MEMA 3 1 0 0 1 ""

 POPS 0 0 0 0 0 0 0 0 1 1 1 1 0 0 0 0

 FLAP 0 3 0

SURF 5

 COMM IOL Posterior

 TYPE STANDARD

 FIMP 

 CURV -5.263157894736841813E-02 0 0 0 0 ""

 HIDE 0 1 0 0 0 0 0 0 0 0

 MIRR 2 0

 SLAB 5

 DISZ 17.274000000000001

 GLAS ___BLANK 2 2 1.3360000000000001 0 0 0 0 0 0 0 

 DIAM 3 1 0 0 1 ""

 MEMA 3 1 0 0 1 ""

 POPS 0 0 0 0 0 0 0 0 1 1 1 1 0 0 0 0

 FLAP 0 3 0

SURF 6

 TYPE STANDARD

 FIMP 

 CURV 0.0 0 0 0 0 ""

 HIDE 0 0 0 0 0 0 0 0 0 0

 MIRR 2 0

 SLAB 8

 DISZ 0

 DIAM 0.040796963281374721 0 0 0 1 ""

 MEMA 5 0 0 0 1 ""

 POPS 0 0 0 0 0 0 0 0 1 1 1 1 0 0 0 0

EFFL 0 1 0 0 0 0 0 0 0 0

LONA 0 0 0 0 0 0 0 0 0 0

TOL TOFF 0 0 0 0 0 0 0 0

MNUM 1 1

MOFF 0 1 "" 0 0 0 1 1 0 0.0 "" 0

 

Regarding the answer to my own comments on IOL constants optimization:

“The purpose of the current study is to investigate the postoperative IOL axial movement, not improve the prediction error of the IOL power calculation formulas.”

Well, why else are we interested in axial IOL movement, if not for right (right lens constant) and stable (good IOL design) refractive results?

Anyways, “we can say that at least 1 month or more after surgery is required to evaluate the postoperative refraction as many studies reported” is a sufficient statement, this doesn’t have to go into an in-depth analysis of every IOL calculation formula.

Response: 

We appreciate this valuable comment, and totally agree with your comment.

I think the statement “the industry standard optical system design” sounds a bit like advertisement, can this be omitted?

Response:

Thank you for this comment. We have deleted the part in the manuscript as recommended.

I figure the refraction lane length was standardized? Which lane length was used?

Response:

Thank you for this comment. CDVA was measured at 5 meters. Therefore, we have added this information in the Method section. Page 7, line 124.

---

## [Editor Report · Decision Letter 2]

9 Aug 2022

Comparison of two one-piece acrylic foldable intraocular lenses: Short-term change in axial movement after cataract surgery and its effect on refraction

PONE-D-22-01418R2

Dear Dr. Maeda,

We’re pleased to inform you that your manuscript has been judged scientifically suitable for publication and will be formally accepted for publication once it meets all outstanding technical requirements.

Kind regards,

Timo Eppig

Academic Editor

PLOS ONE
---

## [Editor Report · Acceptance letter]

22 Aug 2022

PONE-D-22-01418R2 

Comparison of two one-piece acrylic foldable intraocular lenses: Short-term change in axial movement after cataract surgery and its effect on refraction 

Dear Dr. Maeda:

I'm pleased to inform you that your manuscript has been deemed suitable for publication in PLOS ONE. Congratulations! Your manuscript is now with our production department. 

Kind regards, 

on behalf of

Dr. Timo Eppig 

Academic Editor

PLOS ONE